

# Prevalence of food addiction using the Yale-C scale in Mexican children with overweight and obesity

Idalia Cura-Esquivel[1], Juan Ramos-Álvarez[1], Edna Delgado[1] and Airam Regalado-Ceballos[2]

[1] Department of Pediatrics, University Hospital "Dr. José Eleuterio González", Universidad Autónoma de Nuevo León, Monterrey, Nuevo León, Mexico
[2] School of Medicine and University Hospital "Dr. José Eleuterio González", Universidad Autónoma de Nuevo León, Monterrey, Nuevo León, Mexico

## ABSTRACT

**Background**. The prevalence of excess weight and obesity in children has increased significantly worldwide. The concept of food addiction (FA) has been associated with eating-related problems and obesity. Studies on this topic have primarily examined adult samples and little is known about addictive-like eating among Mexican children and adolescents.

**Methods**. We conducted this study to examine the prevalence of FA in a group of 291 overweight and obese children and adolescents using YFAS scale for children (YFAS-C) in Mexico.

**Results**. According to the YFAS-C approximately 14.4% of participants met for FA diagnosis. Forty-two (14.4%) received a FA diagnosis: 14 children and 28 adolescents. The number of FA symptoms in participants who received an FA diagnosis (M = 4.35, SD = 1.07) compared to participants with no FA diagnosis (M = 1.70, SD = 1.53) was significantly higher ($p \leq 0.001$). There were no statistically significant gender differences and the proportion of subjects with an FA diagnosis did not differ by age between children and adolescents. FA is a focus of interest in attempting to explain certain behaviors that may contribute to the development of obesity and explain the failure of the weight loose programs in children.

## INTRODUCTION

The prevalence of excess weight and obesity in children has increased worldwide. A recent meta-analysis reported a prevalence of food addiction (FA) in children of 15% (*Yekaninejad et al., 2021*). In Mexico, the prevalence of excess weight and obesity in the pediatric population has been reported to be 35.6% in children 5 to 11 years old and 38.4% in adolescents 12 to 19 years old, with the northeast of the country being the region with the highest prevalence of childhood obesity (*Romero-Martínez et al., 2019*). However, only one study has evaluated FA in Mexican pediatric population, reporting a prevalence of 20.7% (*Guevara Valtier et al., 2020*). The rise in the prevalence of childhood obesity

Corresponding author
Idalia Cura-Esquivel,
idaliaaracely2008@hotmail.com

might have led to the emergence of serious conditions previously only seen in the adult population, such as diabetes and arterial hypertension.

A contributing factor to the rise in obesity is the availability and wide variety of high-fat, high-sugar foods readily available in small retail stores and supermarkets and to which children have access (*Pineda et al., 2021*; *Pérez-Ferrer et al., 2019*). For some years now, attention has been focused on the biological and physiological effects that the ingestion of certain types of food produces in the individual. It has been suggested that these foods have a potentially addictive effect, similar to the effect produced by drugs and alcohol (*Carlier et al., 2015*; *Hone-Blanchet & Fecteau, 2014*).

The concept of FA has been described for some years, but this had not gained importance until obesity became a pandemic; this concept considers that foods with high levels of sugar and saturated fats produce modifications in behavioral patterns in the brain that create a tendency towards repetitive intake and a dependence similar to substance addiction when trying to give them up or reduce their intake (*Kumar & Kelly, 2017*; *Gearhardt et al., 2011*). This behavior, like other addictions, can create withdrawal symptoms when one tries to stop ingesting them (*Rogers, 2017*; *Volkow, Wise & Baler, 2017*).

The issue of FA in childhood is particularly important since the central nervous system is still developing, so exposure to foods with the potential for addiction can result in unfavorable changes in neurodevelopment, predisposing the child to develop abnormal behavioral patterns concerning food (*Burrows et al., 2017*; *Filgueiras et al., 2019*; *Merlo et al., 2009*).

Although the term FA has not been recognized or defined in the Diagnostic and Statistical Manual of Mental Disorders (DSM-IV), the YFAS, designed by Gearhardt in 2009 (*Gearhardt et al., 2011*; *Gearhardt et al., 2013*), follows the criteria for substance use disorders proposed by the DSM and allows for the objective identification of subjects who exhibit food addictive behaviors. This scale was adapted for use in children and adolescents and validated in multiple populations (YFAS-C) (*Gearhardt et al., 2013*). This scale characterizes FA through elements and criteria that assess the type, quantity, and frequency of food intake, the time spent ingesting it, and its influence on social, occupational, or recreational activities.

Adolescence has been described as a critical period marked by impulsivity and poor inhibition control (*Laurent & Sibold, 2016*; *Lisdahl et al., 2013*) which increases curiosity and the risk of experimenting with too many drugs and alcohol. It has been suggested that unhealthy eating habits and habits with an addictive tendency should also be considered a risk at this age (*Lowe, Morton & Reichelt, 2020*).

This study aimed to evaluate the prevalence of FA in a group of children and adolescents with overweight and obesity and examine the most relevant eating behaviors that occur in this group.

## MATERIALS & METHODS

### Participants

This cross-sectional study was carried out with 291 children and adolescents who attended the Pediatrics department of the "Dr. José Eleuterio González" University Hospital of the

Universidad Autónoma de Nuevo León. The patients chosen for the study attended the consultation for excess weight and obesity, looking for an alternative treatment to lose weight between February 2017 to November 2020. Patients were asked if they wanted to participate in the study by answering the YFAS-C survey without financial compensation. Each child over 8 years of age was given a written informed assent and the parents a written informed consent.

The inclusion criteria were age 5 to 18 years with a BMI ≥ 85th percentile for age and gender according to the Centers for Disease Control and Prevention charts.

## Yale food addiction scale for children

The YFAS-C is a psychometric tool based on the diagnostic criteria for substance dependence described in the Diagnostic and Statistical Manual of Mental Disorders (DSM-IV-TR). It was previous validated for pediatric population in Spanish and Mexican children (*Benítez Brito et al., 2021*; *Moreno et al., 2016*). This scale is based on the seven diagnostic criteria for substance dependence according to the DSM-IV for addictive eating behaviors in the past 12 months; it consists of 25 items, of which 20 relate to 7 dichotomous diagnostic criteria for FA according to the DSM-IV criteria for substance dependence: "Substance ingested in greater quantity and for longer than expected" (Criterion 1); "persistent desire or repeated unsuccessful attempts to cut down consumption" (Criterion 2); "Time invested and effort to obtain, use and recover from the effects" (Criterion 3); "Withdrawal from or reduction of important social, occupational or recreational activities, due to addiction" (Criterion 4); "Persistent use despite knowledge of adverse consequences" (Criterion 5); "Tolerance, marked increase in quantity and decrease in effect" (Criterion 6); "Abstinence" (Criterion 7); An eighth criterion, "Clinically significant decline or distress".

A 5-point Likert scale (0 = never; 4 = always) was applied to the 18 YFAS-C items, and first, a dichotomous scale (yes/no) was used for the seven items.

The score for each criterion is obtained with the sum of its component items and is considered as met (or positive) if it is greater than or equal to 1. The diagnosis of FA is made if participants meet three or more of the seven diagnostic criteria and the distress/disability criterion.

## Data analysis

The SPSS 25.0 statistical software was used to analyze the study data. In terms of demographic patterns, continuous data were stated to be median, and a range was considered a dispersion estimation. Conformity of the data to normal distribution was evaluated using the Kolmogorov–Smirnov test. The $\chi 2$-test was applied to determine if FA was more common in females than males and to compare children and adolescents' prevalence of FA. For continuous variables, such as the number of symptoms between groups with and without FA, the Student's $t$-test was applied. A value of $p \leq 0.05$ was considered statistically significant.

The research ethics committee and the Institution's research committee of the Universidad Autónoma de Nuevo León approved the study procedures (PE16-00005). All participants and their parents were informed about the study, and all gave written informed consent and approval.

## RESULTS

A total of 291 children participated in this study; 163 (56%) were boys and 128 (44%) girls. Participants had a median age of 15 (6–17) years; 95 were children, and 196 were adolescents. The median body mass index (BMI) was 26.5 (17.80–36.50) in boys and 26.2 (18.69–37.10) in girls. According to BMI CDC graphics percentiles, 58% of patients were overweight, and 42% were obese (see Table 1).

According to the YFAS-C scale, forty-two (14.4%) received a FA diagnosis, 14 children (33.3%), and 28 adolescents (66.6%). The mean symptom score was 2.08 ± 1.74.

The most common symptoms were "a persistent desire or repeated unsuccessful attempts to cut down consumption" (46%) followed by "Withdrawal from or reduction of important social, occupational or recreational activities, due to addiction" (38.1%) and "Persistent use despite knowledge of adverse consequences" (37.1%).

When the frequency of symptoms was evaluated by age, the most common symptom in children was the "persistent desire or repeated unsuccessful attempts to cut down consumption" in 57.9%, followed by "Tolerance" (marked increase in amount, marked decrease in effect) in 53.7%. The most common FA symptoms in adolescents were "persistent desire or repeated unsuccessful attempts to cut down consumption" in 40.3% and "Persistent use despite knowledge of adverse consequences" in 39.3%. Children had more symptoms (2.53 ± 1.8) than adolescents (1.87 ± 1.6).

While dividing the group by gender, we observed that boys more frequently (15.3%) met the criteria for the diagnosis of FA than girls (13.3%), but there were no statistically significant gender differences. ($X^2(1) = 0.24$, $p = 0.62$).

The proportion of subjects with an FA diagnosis did not differ by age between children and adolescents ($X^2(1) = .011$, $p = 0.918$).

There was no difference between patients diagnosed with excess weight or obesity and a diagnosis of FA ($X^2(1) = .221$, $p = 0.638$).

## DISCUSSION

Multiple studies have established prevalence and associations with obesity and other eating disorders. This descriptive study examined the prevalence of FA in a group of overweight and obese children and adolescents in a region of northeastern Mexico, a country that reports one of the highest prevalence of obesity worldwide. In this study, the prevalence of FA in children and adolescents with overweight or obesity in the northeastern Mexico region was found to be 14.4%, according to the YFAS-C.

FA is a focus of interest in attempting to explain certain behaviors that may contribute to the development of obesity. The concept of FA suggests a specific eating behavior characterized by excessive and unregulated consumption of energy-dense foods, similar to those behaviors that occur with substance ingestion (*Rogers, 2017*; *Volkow, Wise & Baler, 2017*). In recent years, research into this concept has reported findings suggesting an association between excess weight/obesity and FA.

Cura-Esquivel et al. (2022), *PeerJ*, DOI 10.7717/peerj.13500

**Table 1  Characteristics and comparison between our pediatric population.**

| Symptoms | TOTAL | SEX n (%) | | p | AGE n (%) | | p | WEIGHT n (%) | | p |
|---|---|---|---|---|---|---|---|---|---|---|
| | | Female (n = 128) | Male (n = 163) | | Children (n = 95) | Adolescents (n = 196) | | Overweight (n = 169) | Obesity (n = 122) | |
| Substance ingested in greater quantity and for longer than expected | 41 (14) | 16 (12.5) | 25 (15.3) | 0.49 | 22 (23.2) | 19 (9.7) | 0.002 | 27 (16) | 14 (11.5) | 0.276 |
| Persistent craving or failure to stop ingestion | 134 (46) | 72 (56.3) | 62 (38) | 0.002 | 55 (57.9) | 79 (40.3) | 0.005 | 79 (46.7) | 55 (45.1) | 0.779 |
| Time invested and effort to obtain, use and recover from the effects | 54 (18.5) | 23 (18) | 31 (19) | 0.819 | 28 (29.5) | 26 (13.3) | 0.001 | 35 (20.7) | 19 (15.6) | 0.266 |
| Withdrawal from or reduction of important social, occupational or recreational activities, due to addiction | 111 (38.1) | 48 (37.5) | 63 (38.7) | 0.841 | 46 (48.4) | 65 (33.2) | 0.012 | 62 (36.7) | 49 (40.2) | 0.547 |
| Persistent use despite knowledge of adverse consequences | 108 (37.1) | 50 (39.1) | 58 (35.6) | 0.542 | 31 (32.6) | 77 (39.3) | 0.271 | 68 (40.2) | 40 (32.8) | 0.194 |
| Tolerance, marked increase in quantity and decrease in effect | 88 (30.2) | 41 (32) | 47 (28.8) | 0.556 | 51 (53.7) | 37 (18.9) | <0.001 | 57 (33.7) | 31 (25.4) | 0.127 |
| Abstinence | 72 (24.7) | 37 (28.9) | 35 (21.5) | 0.145 | 28 (29.5) | 44 (22.4) | 0.193 | 45 (26.6) | 27 (22.1) | 0.38 |
| Clinically significant decline or distress | 55 (18.9) | 32 (25) | 23 (14.1) | 0.019 | 24 (25.3) | 31 (15.8) | 0.054 | 33 (19.5) | 22 (18) | 0.748 |
| Food Addiction | 42 (14.4) | 17 (13.3) | 25 (15.3) | 0.62 | 14 (14.7) | 28 (14.3) | 0.918 | 23 (13.6) | 19 (15.6) | 0.638 |
| Mean symptom Score | 2.08 ± 1.74 | 1.94 ± 1.66 | 2.2 ± 1.78 | 0.213 | 2.53 ± 1.80 | 1.87 ± 1.68 | 0.002 | 2.16 ± 1.71 | 1.98 ± 1.79 | 0.381 |

YFAS-C was designed in 2009 by Gearhardt and in 2013 this scale was adapted and validated for use in pediatric populations (*Benítez Brito et al., 2021*; *Moreno et al., 2016*). However, its application is limited, as is the research into this topic in children.

In this study, the prevalence of FA was found to be 14.4%, according to the YFAS-C. This prevalence is in the range reported by other studies in children and adolescents, where prevalence from 7% to 30.7% have been reported (*Yekaninejad et al., 2021*; *Guevara Valtier et al., 2020*; *Naghashpour et al., 2018*; *Schulte et al., 2018*). In her initial study using the modified version of YFAS for a pediatric population, Gearhardt reported a 7.2% prevalence of FA. In contrast, in another study by Tompkins and collaborators (*Tompkins, Laurent & Brock, 2017*) on overweight and obese adolescents, the reported prevalence was 30.7%.

Studies in children have reported that FA has been positively associated with body mass index (BMI), just like the initial study by *Gearhardt et al. (2013)*. In our study, children with obesity showed a similar prevalence of FA than overweight children (15.6% *vs.* 13.6%). This finding is similar to what has been described in adults, where multiple studies report high prevalence of FA in patients seeking treatment for excess weight and/or obesity compared to the general population (*Burmeister et al., 2013*; *Pursey et al., 2014*) These studies report an increased diagnosis of YFAS in obese individuals compared to normal-weight individuals.

Patients included in this study reported as their most common symptom the "persistent desire or repeated unsuccessful attempts to cut down consumption" in 46% of patients with FA and the "important social, occupational or recreational activities given up or reduced" in 38.1%. These symptoms have been commonly reported in adolescent and adult population (*Guevara Valtier et al., 2020*; *Tompkins, Laurent & Brock, 2017*; *Pursey et al., 2014*; *Figueroa-Quiñones & Cjuno, 2018*) . In the group of children, the most frequent symptom was also "persistent desire or repeated unsuccessful attempts to cut down consumption". Children had a higher average of symptoms ($2.5 \pm 1.8$ *vs* $1.87 \pm 1.68$).

There are sensitive periods in human development when an individual may respond in a particular way to a particular environmental stimulus or may more readily adopt abnormal or different behavior patterns compared to other developmental stages (*Knudsen, 2004*). Adolescence is one of these critical periods characterized by increased impulsivity and a decreased ability to control inhibitions, all of which can contribute to the acquisition of psychosocial problems (*Knudsen, 2004*; *Jordan & Andersen, 2017*; *Volkow et al., 2011*). At this stage, the risk of experiencing alcohol and other substance abuse is high, leading to the acquisition of unhealthy eating behaviors and addictive tendencies towards food.

Studies suggest that consumption of certain foods, particularly processed foods rich in saturated fat and high concentrations of sugar, stimulate the mesolimbic reward area in the brain similar to drugs of abuse (*Tompkins, Laurent & Brock, 2017*; *Volkow et al., 2011*; *Gearhardt, 2011*). The onset of drug and alcohol abuse has been described as occurring more frequently in adolescence and early adulthood. In contrast, consumption of potentially addictive foods (*e.g.*, candy, sugar-sweetened beverages) is more likely to occur during early childhood. Although studies in children are scarce, it is possible that FA develops during childhood and adolescence and may be perpetuated into adulthood, and this addictive behavior is one of the theories proposed to explain the high rates of childhood
excess weight and obesity (*Volkow, Wise & Baler, 2017*; *Tompkins, Laurent & Brock, 2017*; *Gearhardt, 2011*).

In this study, the results show that the frequency of FA criteria among children and adolescents was similar (14.7% *vs.* 14.3%), with boys (58.2%) having a higher number of criteria compared to girls (41.8%); however it was not significant. This difference in gender differs from anorexia nervosa and bulimia nervosa where female patients are more affected (*Kinasz et al., 2016*). The mean YFAS-C score was calculated at 2.08 ± 1.74. These data are consistent with those reported in other studies in children where the mean score was 2.06 (*Yekaninejad et al., 2021*; *Kim et al., 2019*) (regardless of symptom) and some adult studies reporting similar scores (2.4 ± 1.8) (*Burmeister et al., 2013*; *Pursey et al., 2014*; *Figueroa-Quiñones & Cjuno, 2018*). Similarly, these results coincide with those obtained in a study conducted in Mexico (*Guevara Valtier et al., 2020*) among high school adolescents where the most common symptom was "Persistent desire or repeated unsuccessful attempt to quit", which was present in 87.7%. Due to these results, the authors have compared this behavior to that of an addiction to any other substance since these behaviors have a biochemical basis where eating, similar to the effect of drugs, can produce pleasure or relaxation, thus creating a dependence on food and resulting in an unsuccessful attempt to reduce food consumption. These repetitive behaviors and the persistent desire to eat in both children and adolescents, despite knowing the consequences similar to substance addiction, are one of the theories proposed to explain the high rates of excess weight and obesity.

Adolescence is a critical period of neurodevelopment, whereby the brain can modify behaviorally in response to the environment. FA follows the same evolution as substance addiction, adolescents with addictive eating behaviors may be at increased risk for FA and the obesity and obesity-related complications will be a problem that will persist into adulthood (*Knudsen, 2004*; *Jordan & Andersen, 2017*; *Dietz, 1994*).

Weaknesses of the study

1. A percentage of children under 8 years of age were included, and to avoid possible comprehension problems, parents were allowed to help their children complete the questionnaire. Parental assistance is considered a bias in this study, as it may affect the interpretation of the questionnaire.
2. The participants in this study comprise a group of children and young people who came to an obesity consultation by their own choice or were motivated to seek treatment by their parents; consequently, the results cannot be interpreted in a generalized manner.
3. This was a cross-sectional study, a longer follow up is suggested to analyze risk factors and evolution of the disease.

## CONCLUSIONS

FA is a focus of interest in trying to explain certain behaviors that may contribute to the development of obesity. The prevalence of FA in this pediatric population with obesity and overweight in a region of northeastern Mexico is similar to that reported worldwide. Mexico ranks first in childhood obesity, so the evaluation of addiction-like eating in

children can contribute to a greater understanding of the eating patterns in childhood and open a new perspective for establish policies for prevention and treatment.

### Funding
The authors received no funding for this work.

### Competing Interests
The authors declare there are no competing interests.

### Author Contributions
- Idalia Cura-Esquivel conceived and designed the experiments, authored or reviewed drafts of the article, and approved the final draft.
- Juan Ramos-Álvarez performed the experiments, authored or reviewed drafts of the article, and approved the final draft.
- Edna Delgado performed the experiments, authored or reviewed drafts of the article, and approved the final draft.
- Airam Regalado-Ceballos analyzed the data, prepared figures and/or tables, and approved the final draft.

### Human Ethics
The following information was supplied relating to ethical approvals (*i.e.*, approving body and any reference numbers):

The Universidad Autónoma de Nuevo León gave Ethical approval to carry out the study within its facilities (Ethical Application Ref: PE16-00005).

### Data Availability
The raw data is available in the Supplemental File.

### Supplemental Information
Supplemental information for this article can be found online at http://dx.doi.org/10.7717/peerj.13500#supplemental-information.

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
