# Peer review of "Prevalence of food addiction using the Yale-C scale in Mexican children with overweight and obesity"

_PeerJ, doi:10.7717/peerj.13500_

## Round 0.1 · original submission · Major Revisions

Thank you for submitting the manuscript to PeerJ. It has been reviewed by experts in the field and we request that you make major revisions before it is processed further.

Please make the required changes, especially those of reviewer 1.

We look forward to hearing from you soon.

Best wishes,

Badicu Georgian, Ph.D

Reviewer 1 ·

Basic reporting

The authors have some interesting data here that contribute to the growing literature on food addiction in children. However, there were some major concerns that need to be address before this manuscript is reading to be published:

-The authors present no hypotheses and do no outline there analyses on the association between food addiction in children and demographic variables anywhere in the introduction. These analyses are also not included in the abstract.

-There was a major meta-analysis on food addiction in children and adolescents recently published. This should be discussed in the intro and discussion.

https://onlinelibrary.wiley.com/doi/full/10.1111/obr.13183

- Given this meta-analysis, the comments about the lack of literature on this topic should be removed or lessened. However, the authors can and should highlight more the lack of research in Mexico (only one other study) in children where obesity rates are so high.

-There is no need for Table 2. This isn't a meta-analysis or structured review.

Experimental design

-The authors spend too much time focusing on the endorsement of specific symptoms and how it differs by demographics. The YFAS-C is a scale that is meant to be either summed up or used as a diagnosis. Each symptom is not meant to stand on their own. Considering most to least endorsed in a table is reasonable, but the authors focus on this too much.

-The authors appear to interpret results that are non-significant. Non-significance should be clearly stated.

-The authors sometimes make statements about food addiction starting in childhood and continuing through adolescence. Given that this study is cross-section, the data can't support this statement. The cross sectional nature of the data should be a limitation and the need for longitudinal research highlighted.

- Information was missing from the methods, including what participants were told the study was about and whether they were compensated. Was a Spanish version of the YFAS-C used? Was it validated? Was it internally consistent?

Validity of the findings

-Through out this paper and in the abstract, the authors need to highlight that this is a sample of only children/adolescents with overweight/obesity who are seeking treatment. This likely inflates the prevalence and reduces the ability to detect differences by BMI and alters the symptoms endorsed. This should be clearly indicated in all interpretation of the findings.

-The discussion begins to feel very repetitive. A discussion of next steps for this literature (specially in Mexican children and adolescents is of importance).

-The lack of gender differences is notable and contrasts with typical eating disorders and should be discussed.

-The findings overall should be discussed in context of the prior meta-analysis.

Additional comments

The authors refer to substance abuse when it should be substance dependence when describing the YFAS-C in the methods

Gearhardt is referred to with a male pronoun, but is a female researcher.

Annotated reviews are not available for download in order to protect the identity of reviewers who chose to remain anonymous.

·

Basic reporting

Comments to the Authors:
In this study, the authors aimed to assess Prevalence of food addiction using the Yale-C scale in
overweight and obese children in Mexico. I have a number of minor and major comments.
1- Please mention to the global prevalence of food addiction in the introduction section before the prevalence of food addiction in Mexico.
2- Please mention "Food Addiction" with the abbreviation "FA" once in the text of the article. Then in the following times, the abbreviated form should be used in the text of the article. Also for the term YFAS in the same way in the text.
3- The word “significant” does not make sense in the discussion section of the article. Please rewrite this sentence.
“The coincidence of our results may be justified by the fact that we included a significant number of adolescents (67%)”.
4- In dissection section, line 219-222, grammatical errors seem to have occurred in this section, and the pronoun his needs to be corrected.
“In his initial study using the modified version of YFAS for a pediatric population, Gearhardt reported a 7.2% prevalence of FA. In contrast, in another study by Tompkins and collaborators (24) on overweight and obese adolescents, the reported prevalence was 30.7%.”
5- The English language should be improved to ensure that an international audience can clearly understand your text. Please review and revise the manuscript for grammatical mistakes.

Experimental design

-

Validity of the findings

-

Additional comments

-

---

## Round 0.2 · accepted · Accept

Thank you for submitting the manuscript to PeerJ. Great improvements were performed in the manuscript. Currently, the article is acceptable for publication.

We look forward to hearing from you soon.

Best wishes,

Badicu Georgian, Ph.D

Reviewer 1 ·

Basic reporting

The manuscript is clear and appropriately reviews the literature. The analytic approaches are appropriate.

Experimental design

The approach is clear and appropriate.

Validity of the findings

The findings are valid and data are appropriate.

Additional comments

The authors have addressed all my concerns